# Which Industrial Sectors Are Affected by Artificial Intelligence? A Bibliometric Analysis of Trends and Perspectives

Lorena Espina-Romero [1,*], José Gregorio Noroño Sánchez [2], Humberto Gutiérrez Hurtado [1], Helga Dworaczek Conde [3], Yessenia Solier Castro [4], Luz Emérita Cervera Cajo [4] and Jose Rio Corredoira [5]

1 Escuela de Postgrado, Universidad San Ignacio de Loyola, Lima 15024, Peru; humberto.gutierrez@epg.usil.pe
2 Facultad de Derecho, Universidad del Sinú "Elías Bechara Zainúm", Monteria 230001, Colombia; josenorono@unisinu.edu.co
3 Programa Maestría en Administración—MBA, Universidad Santo Tomás, Bogota 110231, Colombia; dir.mba@usantotomas.edu.co
4 Escuela de Posgrado, Universidad César Vallejo, Lima 15314, Peru; ssoliercas@ucvvirtual.edu.pe (Y.S.C.); lcerverac@ucv.edu.pe (L.E.C.C.)
5 Escuela de Negocios, Universidad Internacional SEK, Quito 170134, Ecuador; jfrio.mcib@uisek.edu.ec
* Correspondence: lespina@usil.edu.pe

**Abstract:** In recent times, artificial intelligence (AI) has been generating a significant impact in various industry sectors, which implies that companies must be ready to adjust to this promising start and progress in the direction of sustainability. The objective of this paper was to analyze the industrial sectors impacted by artificial intelligence during the period 2018–2022. The methodology consisted of applying a quantitative and bibliometric approach to a collection of 164 manuscripts indexed in Scopus with the help of statistical packages such as RStudio version 4.3.0, VOSviewer version 1.6.19, and Microsoft Excel 365. The results indicate that artificial intelligence is having a growing impact in sectors such as technology, finance, healthcare, the environment, and construction. Geographically, the most impacted sectors are in Europe and Asia, while the least impacted are in the Americas, Africa, and Oceania. It is proposed to conduct future research using AI in power quality (PQ), energy storage systems (ESSs) and hydrogen fuel cell (HFC) systems to contribute, firstly, in the transition to a more sustainable economy, followed by a decrease in dependence on fossil fuels. This research contributes to existing knowledge and paves the way for future exploration of qualitative aspects and emerging trends in the field of artificial intelligence influence in industrial sectors.

**Keywords:** artificial intelligence; industrial sectors; machine learning algorithms; healthcare; sustainability; economic inequality; transportation

## 1. Introduction

So-called artificial intelligence (AI) is a field of computer science that aims to design systems with the ability to execute activities that require human intelligence, including perception, reasoning, learning, and decision-making. This field has progressed in recent years and has become a key technology in many industrial sectors transforming the way we live, work, and interact with the environment around us. Due to its rapid growth, it has the potential to influence all sectors of society, from healthcare [1,2] and finance [3] to transportation [4] and entertainment [5]. Due to its accessibility, industrial sectors and governments lagging with this technology are exploring the potential benefits of applying it in their activities [6].

As researchers seek to understand the full scope of AI's capabilities and potential applications, studies in Scopus on the field have increased in recent years, exploring the technical aspects of artificial intelligence, such as machine learning algorithms [7,8], natural language processing [9,10] and computer vision [11], as well as its ethical and social implications.

Many of these studies have highlighted the potential benefits of artificial intelligence in sectors such as healthcare, where it can improve diagnostics and personalized treatment plans [12], and energy, where it can help companies optimize their use of and reduce operating costs by analyzing substantial amounts of data [13].

One of the sectors affected by artificial intelligence is manufacturing, and this sector is the subject of a report by the consulting firm McKinsey & Company [14], which states that artificial intelligence could increase productivity in the manufacturing sector by 20% by 2030. This is because artificial intelligence can help improve product quality, optimize production processes, and reduce costs. Seconding the previous report, Çankaya and Pekey [15] claim that artificial intelligence can transform manufacturing, ensuring the production of customized and flexible full-range goods and increasing efficiency at all stages of the supply chain.

Healthcare is another sector that has been affected by artificial intelligence, as it can help doctors make more accurate diagnoses and plan personalized treatments. It also helps to analyze massive amounts of medical data and discover patterns and trends undetectable by humans. In the words of Takayanagi [16], AI has the potential to transform medical care, improving quality, efficiency, and accessibility.

Furthermore, in the financial sector, artificial intelligence has also had a significant impact, and it is possible to use it to examine large volumes of information and predict market movements. It can also help detect fraud and prevent money laundering. According to Bhowmik et al. [17], artificial intelligence can transform the finance sector, improving efficiency, reducing risks, and creating new personalized products and services.

However, there are several concerns about the impact of artificial intelligence on both the economy and society. Some studies have focused on the risks this technology could pose to employment, privacy, biased decision-making. Some fear that artificial intelligence could displace human workers, especially in sectors that require repetitive or low-level skills. In an article for Computers in Human Behavior, Bigman et al. [18] argue that AI could create a great deal of wealth but could also increase economic inequality if steps are not taken to redistribute the benefits.

In the literature cited above, several issues related to the impact of artificial intelligence in different industrial sectors are addressed. These include highlighting that artificial intelligence has the potential to transform various sectors, mentioning the potential benefits of the field, and acknowledging that artificial intelligence raises important ethical and social implications. There are specific studies on the impact of AI on individual sectors [19–22], but a complete identification of all affected industrial sectors is lacking. The context of these problems highlights the importance of conducting a bibliometric analysis of the literature in Scopus on the sectors impacted by artificial intelligence during 2018–2022. The objective is to identify the existing knowledge gaps considering bibliometric information but without exploring in detail the qualitative elements. In addition, it will be accompanied by a literature review to understand the opinions and perspectives of the authors involved in the selected papers. By doing so, this research will contribute to a more complete understanding of the impact of artificial intelligence in various industrial sectors and will provide relevant information on the advances, challenges, and opportunities in this field in the context of different industries. From this perspective, the present research posed the following question: What are the industrial sectors impacted by artificial intelligence?

To answer this research question, this study continues with a structure that includes the methodology used, consisting of the study design, bibliometric data collection, analysis, and visualization of results, as well as their interpretation. It is followed by the results and discussion section, which is divided into sections that present relevant information about this research, such as the study period, the sources used, the number of documents selected, and their annual growth rate. It also analyzes the documents registered by year and country, as well as the most relevant sources in the field of artificial intelligence. The industrial sectors affected and the key areas for future studies are identified. Finally, conclusions and future perspectives are presented.

## 2. Methodology

The quantitative methodology of this manuscript is based on bibliometric examination [23] but with the five steps suggested in Zupic and Čater's research [24]. These steps include "study design", "bibliometric data collection", "analysis", "visualization" and the "interpretation".

### 2.1. Study Design

This bibliometric study posed a research question in the introduction section and, to answer it, line graphs with Microsoft Excel markers and tables were used to display changes in the data or to illustrate comparisons between elements (main information, publications and citation average per year, publications and total citations per country, most cited papers, relevant sources, and the trend of industry sectors). To show the terms that allow us to infer the industrial sectors impacted by artificial intelligence, a bibliometric methodology called "keyword co-occurrence analysis" with "overlay visualization" was used. To extract future research topics, a "keyword co-occurrence analysis"-type bibliometric methodology with "network visualization" was used to display keywords with minimal co-occurrence.

### 2.2. Bibliometric Data Collection

The Scopus database was used because of its extensive catalog of articles, many of which are indexed in other databases. Searches were performed using the following terms in the TITLE-ABS-KEY field: "Artificial Intelligence" or "AI" or "Cognitive Computing" or "Machine Learning" or "Intelligent Automation" or "Machine Intelligence" or "Robotic Intelligence" or "Smart Computing" or "Deep Learning" or "Digital Intelligence" or "Neural Networks" or "Expert Systems" or "Computer Vision" or "Knowledge Engineering" or "Natural Language Processing (NLP)" or "Autonomous Systems" and "industrial subsectors" or "industry sectors" or "Manufacturing segments" or "Industrial divisions" or "Production categories" or "Business branches" or "Economic sectors" or "Commercial spheres" or "Trade categories" or "Product categories". These terms were selected after a literature review, as they were considered the most relevant to artificial intelligence in industrial sectors. A total of 787 documents were obtained in the initial search, which were narrowed down to 233 open access records and a specific period (2018–2022) to obtain more recent literature. reviews, book chapters, books, data paper, errata, and short-survey-type documents were excluded as studies with outcomes were required. This resulted in 164 records selected for analysis (See Figure 1).

The search key generated was the following: "TITLE-ABS-KEY ("Artificial Intelligence" OR "AI" OR "Cognitive Computing" OR "Machine Learning" OR "Intelligent Automation" OR "Machine Intelligence" OR "Robotic Intelligence" OR "Smart Computing" OR "Deep Learning" OR "Digital Intelligence" OR "Neural Networks" OR "Expert Systems" OR "Computer Vision" OR "Knowledge Engineering" OR "Natural Language Processing (NLP)" OR "Autonomous Systems" AND "industrial subsectors" OR "industry sectors" OR "Manufacturing segments" OR "Industrial divisions" OR "Production categories" OR "Business branches" OR "Economic sectors" OR "Commercial spheres" OR "Trade categories" OR "Product categories") AND (LIMIT-TO (OA, "all")) AND (LIMIT-TO (PUBYEAR, 2022) OR LIMIT-TO (PUBYEAR, 2021) OR LIMIT-TO (PUBYEAR, 2020) OR LIMIT-TO (PUBYEAR, 2019) OR LIMIT-TO (PUBYEAR, 2018)) AND (LIMIT-TO (DOCTYPE, "ar") OR LIMIT-TO (DOCTYPE, "cp"))".

### 2.3. Analysis

At this stage, the uploaded information was converted to ensure its quality and usefulness. The data, derived from Scopus and in CSV format, were uploaded to various tools such as VOSviewer, RStudio, and Microsoft Excel 365.

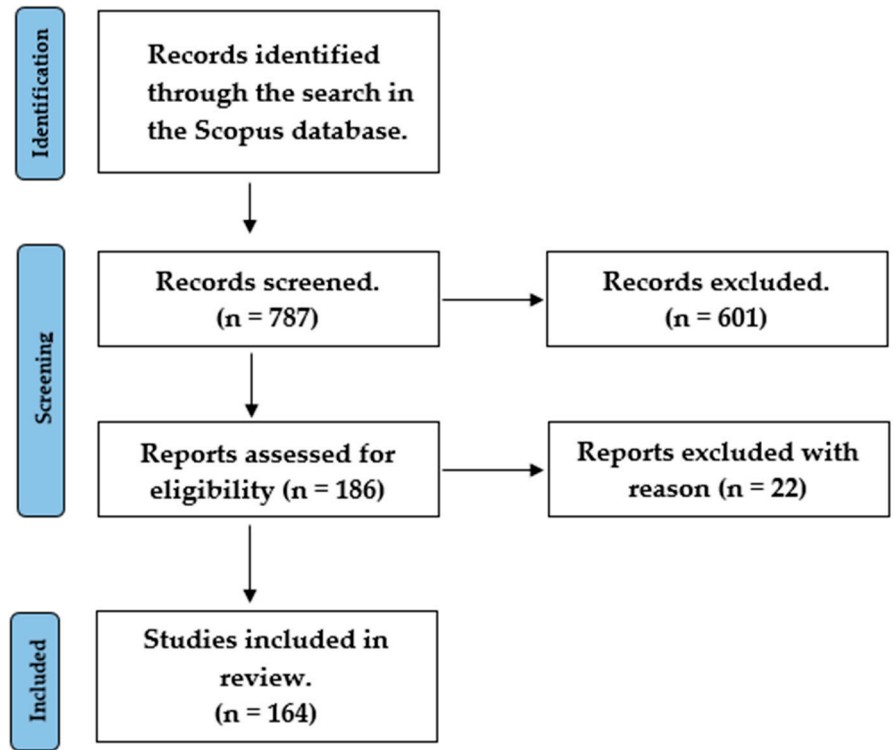

**Figure 1.** Process of sample selection: PRISMA method.

### 2.4. Visualization

An analysis was conducted using line graphs with markers in Microsoft Excel 365 for the purpose of displaying annual document production and trends across industry sectors. An analysis was performed using tables to show main information, publications and citation average per year, publications and total citations per country, most cited papers, and relevant sources. In addition, an analysis was performed using VOSviewer software (version 1.6.19), which employed "keyword co-occurrence analysis" to visualize industries impacted by artificial intelligence and topics that could be the subject of future research.

### 2.5. Interpretation

In this phase, the results obtained according to the research question of this study, which focused on providing key information about artificial intelligence, the industries impacted by it and the topics that should be considered in the research agenda, were presented and analyzed. The inferences were made in accordance with the specific objectives established for the creation of the final document.

### 3. Results and Discussion

### 3.1. Main Information

Table 1 presents the main data of studies related to "artificial intelligence and industrial sectors" conducted in the period 2018–2022.

A total of 125 sources were used to conduct this study, including scientific journals. The total number of papers selected in this case is 164. The annual growth rate of artificial intelligence research in industrial sectors was 47.35%, i.e., research on the topic was productive, and the number of manuscripts and journals worked on increased significantly each year. The average age of the manuscripts worked was 2.23 years, i.e., this study focused on recent papers; therefore, up-to-date information was used. On average, each paper received 10.65 citations, implying that this study was based on highly cited and high-quality papers.

**Table 1.** Main Information.

| Description | Results |
|---|---|
| "Timespan" | 2018:2022 |
| "Sources" | 125 |
| "Documents" | 164 |
| "Annual Growth Rate %" | 47.35 |
| "Document Average Age" | 2.23 |
| "Average citations per doc" | 10.65 |
| "References" | 7746 |
| "Keywords Plus (ID)" | 1219 |
| "Author's Keywords (DE)" | 657 |
| "Authors" | 722 |
| "International co-authorships %" | 29.27 |
| "Article" | 124 |
| "Conference Paper" | 40 |

The total number of references cited in the 164 documents is 7746, suggesting that a considerable amount of reference information was used to develop this research. A total of 1219 pus keywords were used, implying that a detailed search for valuable information was conducted. A total of 657 author keywords were identified to conduct this study; this implies that the authors were up to date with the study topic and provided specific keywords to ensure a meaningful information search. The total number of authors who participated in the 164 papers is 722, suggesting that they made a collaborative effort when researching the topic in question.

The percentage of international co-authorship is 29.27%, allows us to infer that the development of the 164 documents was an international effort, i.e., the authors come from different countries. Of the total number of documents, 124 are articles and 40 are conference papers, suggesting that information has been selected from primary sources. This may help to better understand the state of knowledge and the evolution of the artificial intelligence field in industrial sectors.

*3.2. Registered Documents by Year and Average Citation Rate*

The data presented in Figure 2 indicate the number of papers published during 2018–2022 and the average number of citations received for each year.

It can be noted that the number of studies has increased each year, from 14 in 2018 [5,25–29], 21 in 2019 [16,30–34], 20 in 2020 [35–40], and 43 in 2021 [18,41–45] to 66 in 2022 [1,46–50], i.e., an annual growth rate of 47.35%. Despite all this, the average number of citations per manuscript has decreased from 6.46 in 2018 to 2.29 in 2022, indicating that more recently published research is receiving less attention and recognition as opposed to research published in previous years. Similarly, it can be noted that despite the growth in the number of published papers in 2021, the average number of citations remained constant compared to the previous year.

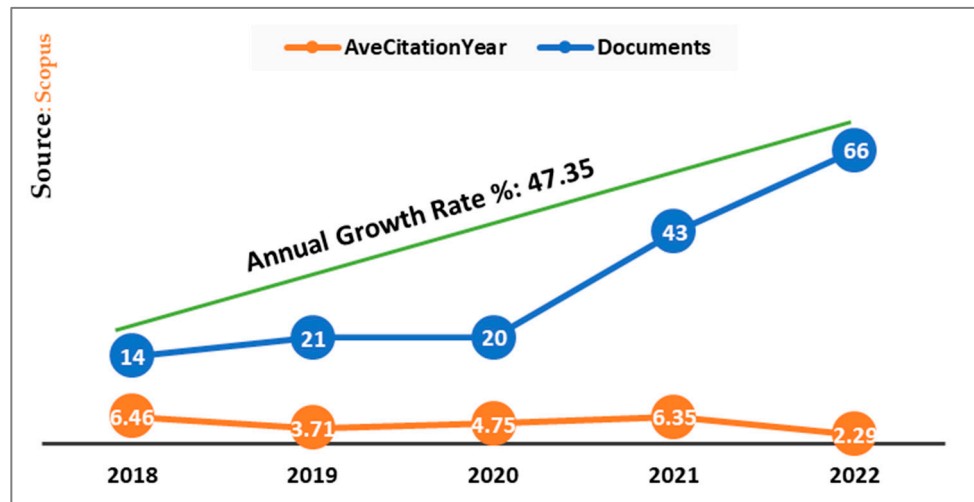

**Figure 2.** Documents per year and average citations.

### 3.3. Registered Documents by Country and Their Total Citations

Figure 3 provides the data of the ten countries with the highest number of documents and their total citations. These countries are China [42,51–55], United States [32,44,56–59], United Kingdom [31,34,60–63], Germany [26,35,64–67], India [10,41,68–71], Italy [31,52,72–75], Spain [31,76,77], Canada [47,68,78], Indonesia [79–81], and Australia [82–84]. The country with the highest number of total citations is the United Kingdom, with 205 citations, followed by Spain with 103 citations and then Italy with 83 citations. China has the highest number of papers with 28, followed by the United States with 27, and the United Kingdom with 16. The countries with the fewest papers are Australia with seven and Canada and Indonesia with eight each.

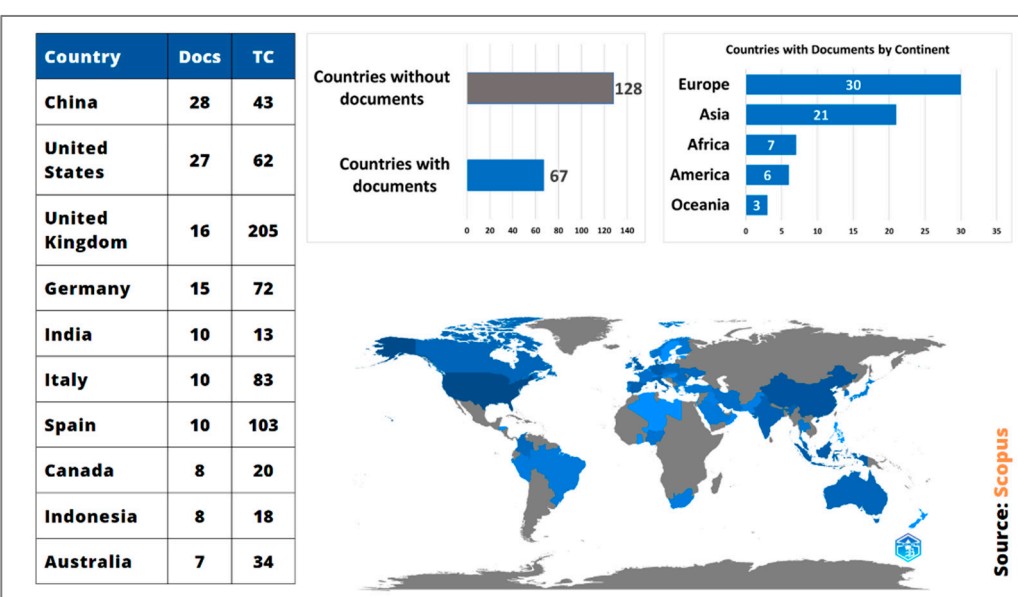

**Figure 3.** Publications and total citations by country.

Of the 195 countries in the world, only 67 countries (34.36%) managed to register publications in Scopus on the variables in question, i.e., 128 countries (65.64%) did not register studies on this platform. Figure 3 also shows that Europe is the continent with the largest number of countries with documents, with a total of 30, representing a percentage of 44.78% of the 67 countries with studies on the subject. Asia follows in second place, with 21 countries, equivalent to a percentage of 31.34%. The Americas and Africa have fewer

countries with documents compared to Europe and Asia, with six and seven countries respectively, and percentages of 8.96% and 10.45%. Finally, Oceania is the continent with the fewest countries with documents, with a total of three countries and a percentage of 4.48%.

### 3.4. Most Relevant Sources

Table 2 presents information on the number of documents (Docs), h-index (h_index), total citations (TCs) and year of start of publication (PY_start) of the ten relevant sources in the field of AI in industrial sectors.

**Table 2.** Relevant sources.

| Relevant Sources | Docs | h_index | TCs | PY_start |
|---|---|---|---|---|
| *"Sustainability* **(Switzerland)"** | 8 | 4 | 100 | 2018 |
| *"IEEE Access"* | 7 | 3 | 44 | 2019 |
| *"Lecture Notes in Computer Science (Including Subseries Lecture Notes in Artificial Intelligence and Lecture Notes in Bioinformatics)"* | 6 | 2 | 9 | 2019 |
| *"Applied Sciences* **(Switzerland)"** | 4 | 3 | 26 | 2019 |
| *"Journal Of Physics: Conference Series"* | 4 | 2 | 12 | 2018 |
| *"Procedia CIRP"* | 3 | 2 | 85 | 2018 |
| *"Sensors"* | 3 | 1 | 6 | 2021 |
| *"AIP Conference Proceedings"* | 2 | 0 | 0 | 2021 |
| *"Electronics* **(Switzerland)"** | 2 | 1 | 3 | 2021 |
| *"Energies"* | 2 | 1 | 2 | 2021 |

From the contents of Table 2, it can be noticed that the source with the highest number of publications is *"Sustainability (Switzerland)"* with eight papers published since 2018 [2,4,40,56,75,85–87], followed by *"IEEE Access"* with seven papers published since 2019 [13,32,63,69,88–90]. The highest h-index is 4 for the source *"Sustainability (Switzerland)"*, indicating that at least four of its papers have been cited at least four times each. The highest CT is 100 for *"Sustainability (Switzerland)"*, followed closely by *"Procedia CIRP"* with 85 citations.

A substantial portion of the sources in the table have a low h-index and total citations, suggesting that their publications have been cited in the literature infrequently. The sources *"Sensors"* [19,91,92], *"AIP Conference Proceedings"* [48,49], *"Electronics (Switzerland)"* [93,94], and *"Energies"* [95,96] have a limited number of articles and citations, suggesting that they are less relevant sources than the others.

### 3.5. Most Relevant Documents

Table 3 shows information on the number of citations received by the 10 most relevant scientific papers, together with the author and the DOI corresponding to each of them. The papers are ordered according to the total number of citations they have received.

Author Jadhav S. in 2018 has the most citations with a total of 160, indicating that his paper has been very influential and has been cited by many authors. The second author with the second highest number of citations is Muhammad L. J. in 2021 with 129 citations, followed by Langley D. J. in 2021 with 105 citations. It can also be seen that the most cited papers were published in 2018, 2019, and 2021, which allows us to infer that they are new enough to the topic and are gaining interest and discussion among researchers.

**Table 3.** Most cited documents.

| Document Prepared by | DOI | Total Citations |
|---|---|---|
| Jadhav S, 2018 [60] | 10.1016/j.asoc.2018.04.033 | 160 |
| Muhammad L J, 2021 [68] | 10.1007/s42979-020-00394-7 | 129 |
| Langley D J, 2021 [34] | 10.1016/j.jbusres.2019.12.035 | 105 |
| Taherei Ghazvinei P, 2018 [26] | 10.1080/19942060.2018.1526119 | 88 |
| Nilashi M, 2019 [82] | 10.1016/j.jclepro.2019.01.012 | 73 |
| Hajek P, 2020 [97] | 10.1007/s00521-020-04757-2 | 61 |
| Awan M J, 2021 [61] | 10.32604/iasc.2021.014216 | 53 |
| Saura J R, 2018 [56] | 10.3390/su10093016 | 48 |
| Ogorodnyk O, 2018 [98] | 10.1016/j.procir.2017.12.229 | 46 |
| Huang C, 2019 [99] | 10.1145/3292500.3330790 | 45 |

*3.6. Industry Sectors Impacted by AI*

Figure 4 illustrates AI-related terms in various industry sectors over the period 2018 to 2022. VOSviewer software (version 1.6.19) was used to create the figure, employing a keyword co-occurrence analysis, with a fractional count and a minimum of one occurrence of a keyword.

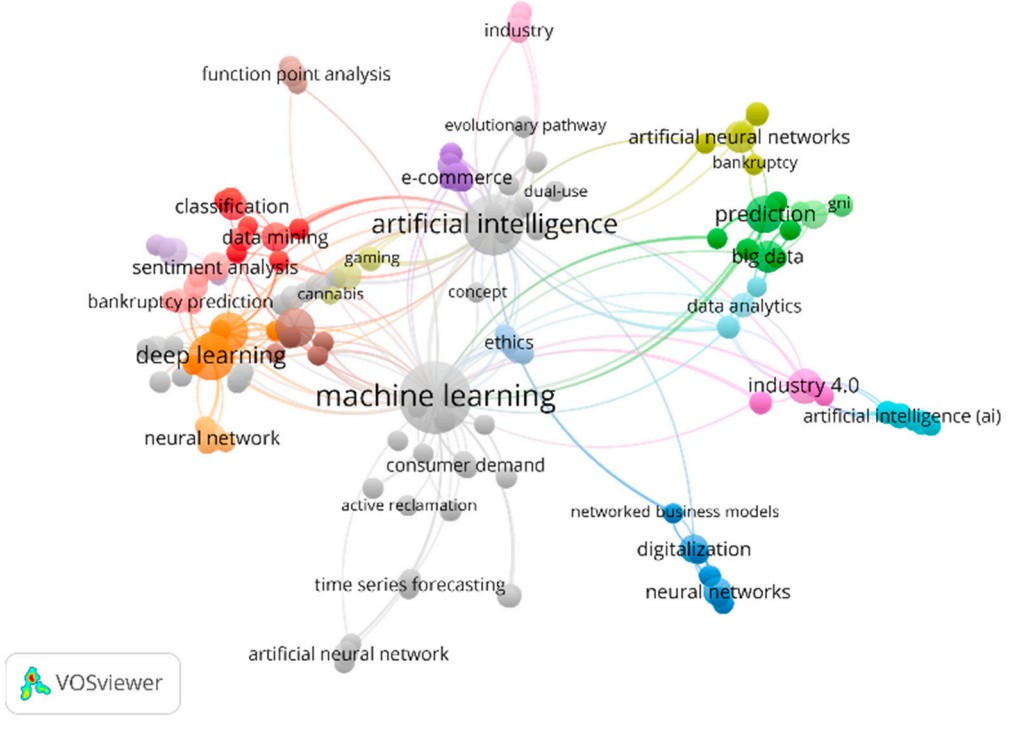

**Figure 4.** Keywords linked to artificial intelligence and industry sectors.

All identified terms were classified in Table 4 according to their frequency of occurrences, their average annual publication (avg. pub. year) and the corresponding industry sector. A detailed analysis of the impacted sectors is provided below the table.

**Table 4.** Terms related to artificial intelligence and industry sectors.

| Label/Term | Weight (Occurrences) | Score (Avg. Pub. Year) | Industry Sector |
|---|---|---|---|
| artificial wolf-pack algorithm | 1 | 2018.00 | Technological |
| business leadership | 1 | 2018.00 | Business |
| digital agriculture | 1 | 2018.00 | Agriculture |
| environmental factors | 1 | 2018.00 | Environmental |
| environmental management | 1 | 2018.00 | Environmental |
| eWOM | 1 | 2018.00 | Business |
| evolutionary pathway | 1 | 2018.50 | Investigation |
| commercial content | 1 | 2019.00 | Business |
| data analytics | 2 | 2019.00 | Technological |
| digital capability | 1 | 2019.00 | Technological |
| food and agricultural ethics | 1 | 2019.00 | Agriculture |
| generative adversarial networks | 2 | 2019.50 | Technological |
| big data | 4 | 2019.75 | Technological |
| prediction | 6 | 2019.83 | Technological |
| accuracy | 1 | 2020.00 | Technological |
| advanced manufacturing | 1 | 2020.00 | Manufacturing |
| applications | 1 | 2020.00 | Technological |
| assembly | 1 | 2020.00 | Manufacturing |
| CIO | 1 | 2020.00 | Business |
| clustering | 1 | 2020.00 | Technological |
| data management | 1 | 2020.00 | Technological |
| digital migration | 1 | 2020.00 | Public services |
| exploratory projection pursuit | 1 | 2020.00 | Finance |
| GDP | 3 | 2020.00 | Finance |
| genetic algorithm in wrapper | 1 | 2020.00 | Technological |
| human capital | 1 | 2020.00 | Business |
| industrial big data | 1 | 2020.00 | Manufacturing |
| industrial internet of things | 1 | 2020.00 | Manufacturing |
| object recognition | 2 | 2020.00 | Technological |
| e-commerce | 3 | 2020.33 | Business |
| hydrogen fuel cell (HFC) | 1 | 2020.50 | Energy |
| personalization | 2 | 2020.50 | Business |
| sentiment analysis | 3 | 2020.67 | Technological |
| industry 4.0 | 5 | 2020.80 | Manufacturing |
| machine learning | 33 | 2020.88 | Technological |
| acceleration signal | 1 | 2021.00 | Education |
| air purifier development | 1 | 2021.00 | Health |
| black Friday sales | 1 | 2021.00 | Business |
| cellular agriculture | 1 | 2021.00 | Agriculture |
| cloud | 1 | 2021.00 | Technological |
| co-creation | 1 | 2021.00 | Business |

**Table 4.** *Cont.*

| Label/Term | Weight (Occurrences) | Score (Avg. Pub. Year) | Industry Sector |
|---|---|---|---|
| commuting | 1 | 2021.00 | Transportation |
| content automation | 1 | 2021.00 | Business |
| control techniques | 1 | 2021.00 | Education |
| correlation and regression analysis | 1 | 2021.00 | Technological |
| deep learning | 11 | 2021.00 | Technological |
| democracy | 1 | 2021.00 | Public services |
| digital specialists | 1 | 2021.00 | Business |
| forestry | 1 | 2021.00 | Forestry |
| gaming | 1 | 2021.00 | Entertainment |
| higher vocational education | 1 | 2021.00 | Education |
| housing price | 1 | 2021.00 | Business |
| latent Dirichlet allocation | 2 | 2021.00 | Technological |
| neural networks | 3 | 2021.00 | Technological |
| pandemic | 2 | 2021.00 | Health |
| social media | 3 | 2021.00 | Technological |
| natural language processing | 7 | 2021.14 | Technological |
| data mining | 3 | 2021.33 | Technological |
| artificial neural networks | 4 | 2021.50 | Technological |
| consumer demand | 2 | 2021.50 | Business |
| information gain | 1 | 2021.50 | Technological |
| time series forecasting | 2 | 2021.50 | Finance |
| COVID-19 | 7 | 2021.57 | Health |
| online reviews | 3 | 2021.67 | Business |
| ability to learn | 1 | 2022.00 | Education |
| AI adoption challenges | 1 | 2022.00 | Business |
| AI opportunities | 1 | 2022.00 | Business |
| AI-based systems | 1 | 2022.00 | Technological |
| anthropomorphism | 1 | 2022.00 | Investigation |
| bankruptcy prediction | 2 | 2022.00 | Finance |
| c45 | 1 | 2022.00 | Technological |
| c89 | 1 | 2022.00 | Public services |
| challenges of distribution network system (DNS) | 1 | 2022.00 | Technological |
| city manager | 1 | 2022.00 | Public services |
| construction ecosystem | 1 | 2022.00 | Construction |
| construction technology | 1 | 2022.00 | Construction |
| consumer response | 1 | 2022.00 | Business |
| COVID-19 response | 1 | 2022.00 | Health |
| credit scoring | 1 | 2022.00 | Finance |
| d12 | 1 | 2022.00 | Entertainment |

**Table 4.** *Cont.*

| Label/Term | Weight (Occurrences) | Score (Avg. Pub. Year) | Industry Sector |
|---|---|---|---|
| diagnosis | 1 | 2022.00 | Health |
| distribution static synchronous compensator (d-statcom) | 1 | 2022.00 | Energy |
| energy storages system (ESS) | 1 | 2022.00 | Energy |
| energy-related Co2 emissions | 1 | 2022.00 | Environmental |
| function point analysis | 2 | 2022.00 | Technological |
| Heroku | 1 | 2022.00 | Technological |
| information trustworthiness | 1 | 2022.00 | Technological |
| local government | 2 | 2022.00 | Public services |
| LSTM | 2 | 2022.00 | Technological |
| software effort estimation | 2 | 2022.00 | Technological |
| technology adoption | 2 | 2022.00 | Technological |

Table 4 allows us to extract 15 industrial sectors impacted by artificial intelligence. These sectors are technological, business, finance, health, manufacturing, public service, education, agriculture, energy, environmental, construction, entertainment, investigation, forestry, and transportation.

First, the technology industry is the most impacted by artificial intelligence, as it has the highest number of mentions in all years of the period considered. Moreover, there is an increasing trend in the impact of artificial intelligence on this sector over time. A study related to this sector, was developed by Kar et al. [69], where they state that the Industrial Internet of Things (IIoT) and digital technologies have evolved rapidly, creating new job profiles and generating a challenge in the availability of skills. Another study linked to this sector was developed by Habib and Hamadneh [85], where they point out that e-commerce has experienced spectacular growth globally due to the COVID-19 pandemic and the advance of mobile technology, which has led to rapid consumer adoption.

The business sector also has a considerable upward trending impact from artificial intelligence. From this sector, the study by Gupta et al. [27] was identified, where they analyzed how social media usage and sentiment vary across different industry sectors and geographic regions using specific data from Twitter and the UK Standard Industrial Classification Code. In addition, Awan et al. [61] conducted a study to help retailers design personalized offers and promotions using a big data framework that ensures massive sales volumes with more efficient models using Black Friday sales data taken from the Kaggle website.

Next, the financial sector has experienced the largest increase in the impact of artificial intelligence in recent years, with a significant increase in 2020, 2021, and 2022. A study linked to this sector is that of Jadhav et al. [60], which investigates feature selection for credit score improvement, which is crucial in assessing the creditworthiness of individuals and companies, employing machine learning techniques and feature selection based on genetic algorithms to improve the performance of credit rating models. In parallel, Gavurova et al. [1] wrote about the importance of assessing the financial health of companies to predict their future development and improve their financial performance and competitiveness. The aim of the study was to anticipate the bankruptcy of companies in the engineering and automotive sectors in the Slovak Republic with the help of a multilayer neural network system and logistic regression. The findings showed that the financial indicators QR, ROS, PC/S, and NWC/A mitigate the risk of bankruptcy.

The healthcare sector has also experienced a growing impact of artificial intelligence over time, with mentions in the years 2021 and 2022. From this sector, studies such as

Muhammad et al. [68] were identified, in which machine learning models were developed using labeled epidemiological data from patients with COVID-19 in Mexico. Several learning algorithms were used, and the performance of each model was evaluated. The decision tree model had the highest accuracy, and the support vector machine model had the highest sensitivity. Another study identified was written by Cleland et al. [62], where they examined public health policy development using data science and machine learning methods. The benchmark for antidepressant medication in Northern Ireland was reviewed, and a view was put forward suggesting that the relationship between antidepressant use and economic deprivation is mediated by the impact of depression.

The manufacturing sector has been impacted by artificial intelligence during 2020 and has maintained a steady number of mentions throughout that year [39,75,100]. The public services sector has been mentioned in some years but has experienced an increase in artificial intelligence impact in recent years [82,84,101].

Regarding education [40,71], the agriculture sector [44,72], and the energy sector [42,77], all have seen a varying number of artificial intelligence mentions over time. The environmental sector [26,56] and construction [83,102] have seen an increase in artificial intelligence impact in recent years.

On the other hand, in the entertainment industry [2,5] and investigation [50,103] a variable number of artificial intelligence mentions is observed, although the impact in these sectors seems to be increasing in recent years. The forestry industry [76,104] and transportation [4,49] have been mentioned only in the year 2021 of the period considered, with a variable amount of artificial intelligence impact.

Overall, it is observed that artificial intelligence is increasingly impacting a variety of industry sectors and is anticipated to continue to be a trend in the future. The trend of the fifteen sectors impacted by artificial intelligence during 2018–2022 is visualized in Figure 5.

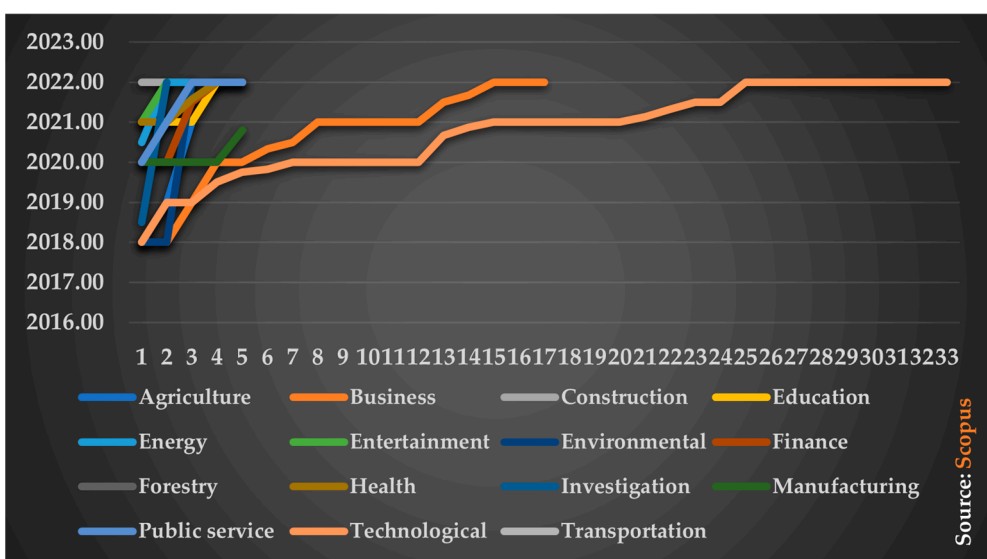

**Figure 5.** Trends in industrial sectors impacted by artificial intelligence.

### 3.7. Recommended Topics for Future Research

To establish a research agenda for the future, 657 author keywords were entered into the statistical program VOSviewer to perform the "keyword cooccurrence analysis" based on the "strength of association" method. The selection of keywords was made considering those with the lowest co-occurrence (*n* = 1) related to artificial intelligence in industrial sectors. From this analysis, three relevant themes were identified (Figure 6) that could be the subject of future study. These topics are power quality (PQ), energy storage systems (ESSs), and hydrogen fuel cells (HFCs).

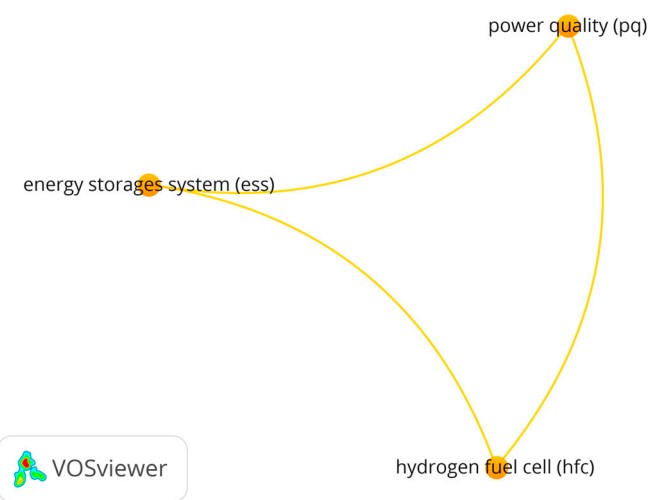

**Figure 6.** Topics for future research.

- Power quality (PQ) studies focus on the use of artificial intelligence to detect power disturbances, analyze voltage, and current variation, regulate power quality, and find technical solutions to improve power quality [96].
- Energy storage system (ESS) research focuses on the use of artificial intelligence to evaluate the efficiency and cost-effectiveness of different energy storage technologies as well as to identify specific applications for these systems [28].
- Hydrogen fuel cell (HFC) studies refer to the use of artificial intelligence to evaluate the efficiency and cost-effectiveness of hydrogen fuel cells in distinct types of technologies as well as to identify specific applications [105].

The sectors that could be impacted by these three AI-related topics would be manufacturing, telecommunications, renewable energy, electric mobility, transportation, and electronics.

## 4. Conclusions and Limitations

This study provides valuable data on the impact of artificial intelligence in various industrial sectors and highlights the novelty and importance of its findings, contributes to the existing knowledge on the subject, and acknowledges its limitations.

The novelty of this study lies in its comprehensive analysis of the literature in Scopus, examining the sectors impacted by artificial intelligence during the period 2018–2022. Unlike previous studies that focused on specific sectors, this study aimed to identify all industrial sectors impacted by artificial intelligence by employing a bibliometric analysis to explore trends, influential authors, sources, and papers, shedding light on the research landscape in this domain.

The study contributes to existing knowledge by providing a detailed overview of the industrial sectors affected by artificial intelligence by identifying key sectors such as technology, business, finance, health, the environment, and construction, and highlights the potential benefits of artificial intelligence in improving productivity, quality, efficiency, and accessibility in these sectors. The findings also emphasize the transformative power of this technology in enabling personalization, flexibility, and optimization across the supply chain.

However, it is important to recognize the limitations of this study. The analysis is based on the Scopus database, and, while it provides a wide range of articles, it may not capture the entirety of research in this field. In addition, the study focuses on the period 2018–2022, and rapid advances in artificial intelligence may have led to new developments beyond this time limit. In addition, the study is based on bibliometric data and does not delve into the qualitative aspects of the impact of artificial intelligence on industrial sectors.

Future research can build on these findings to delve deeper into the qualitative aspects and emerging trends of the impact of artificial intelligence in industrial sectors.

It is recommended to conduct studies where artificial intelligence can be used to improve the monitoring, control, diagnosis, optimization and automation of power quality (PQ), energy storage systems (ESSs) and hydrogen fuel cell (HFC) systems and to analyze substantial amounts of data to improve their efficiency and performance.

**Author Contributions:** Conceptualization, L.E.-R.; methodology, J.G.N.S. and H.G.H.; software, H.D.C.; validation, Y.S.C., L.E.C.C. and J.R.C.; formal analysis, J.G.N.S.; investigation, L.E.-R.; resources, H.G.H.; data curation, H.D.C.; writing—original draft preparation, Y.S.C.; writing—review and editing, L.E.C.C.; visualization, J.R.C.; supervision, L.E.-R.; project administration, J.G.N.S. All authors have read and agreed to the published version of the manuscript.

**Funding:** This research received no external funding.

**Institutional Review Board Statement:** Not applicable.

**Informed Consent Statement:** Not applicable.

**Data Availability Statement:** This study utilized a bibliometric approach, and the data used were generated from the Scopus database.

**Conflicts of Interest:** The authors declare no conflict of interest.

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
