# Peer review of "Which Industrial Sectors Are Affected by Artificial Intelligence? A Bibliometric Analysis of Trends and Perspectives"

_sustainability, doi:10.3390/su151612176_

Round 1

Reviewer 1 Report

Report
The overall contribution is good. I suggest a major revision. The following are my observations:

Strengths:
1. This is an important research topic.
2. The structure of this review paper is comprehensive.

Suggestion:

Initially, names and the affiliations of this paper are missed. Please check twice before submitting the revision.

Weaknesses:
1. There should be some taxonomy that present the hierarchy of the review, which can explain the the current problem involving in this topic. It should have a clear flow of all processes i.e., which process addresses first, then the next process and so on.
2. Background and related work: Generally, the section provides many references, but it remains unclear which problems are being highlighted in the cited literature, where the prior art falls short of the authors expectations and requirements for an AI/ML.
3. The novelty and contribution of this paper are limited, and the disadvantages of the review taxonomy were not clearly discussed as compared to other state-of-the-art review articles.
4. Please check abbreviations in this paper. Many abbreviations in the paper are referenced only once or twice, the authors are encouraged to reduce their use where possible to aid in legibility.
5. Please check contribution section of this paper, there is a need to highlight novelty of this review.
6. There are lot of works that have used AI/ML. Authors should include these works in related studies by including a table mentioning the pros and cons of these works.
7. There are many more. Authors should do a thorough literature survey and include all such papers in related studies.
8. Authors should proofread entire manuscript for grammatical mistakes.

Minor english editing is required before submitting revision.

Reviewer 2 Report

This article is well presented. The reviewer recommends the authors to consider the blow recommendation.

Line 9-10: the declaration is abrupt and does not flow well.

The introduction session could benefit from more detailed examples. 

Figure 5 looks confusing

Reviewer 3 Report

1. The information of the author is missing, the author contribution is missing.

2. The figures should be polished carefully.

3. The introduction should give the structure of your study.

The English is good.

Reviewer 4 Report

This paper presents

Lately, Artificial Intelligence (AI) is impacting several industrial sectors, therefore, companies must be prepared to adapt to this new beginning and move towards sustainability. The objective of this paper was to analyze the industrial sectors impacted by Artificial Intelligence during the period 2018-2022. The methodology consisted of applying a quantitative and bibliometric approach to a collection of 164 manuscripts indexed in Scopus with the help of statistical packages such as RStudio, VOSviewer and Microsoft Excel. The results indicate that Artificial Intelligence is having a growing impact in sectors such as technology, finance, healthcare, environment, and construction. Geographically, the most impacted sectors are in Europe and Asia, while the least impacted are in the Americas, Africa, and Oceania. It is proposed to carry out future research using AI in Power Quality (PQ), Energy Storage Systems (ESS) and Hydrogen Fuel Cell (HFC) systems to contribute, firstly, in the transition to a more sustainable economy, followed by a decrease in dependence on fossil fuels.

I do not recommend this paper for publication in this journal. The paper is a more theoretical type and its needs more improvement in terms of adding figures and data.

Reviewer 5 Report

Manuscript “Which Industrial Sectors Are Affected by Artificial Intelligence? A Bibliometric Analysis of Trends and Perspectives is a good and informative study but had some minor corrections. Below are some comments/suggestion for the authors to improve its quality:

  1. Language should be improvised.
  2. In Abstract section methodology of study should be included.
  3. Also abstract section should include some conclusive statements.
  4. Introduction section must include the need and significance of study.
  5. Introduction section must include some information on the importance of flavonoids to be taken for treatment.
  6. Main manuscript also include methodology section; clearly stating inclusion and exclusion criteria.
  7. Conclusions: The conclusions are too general, format according to future aspects. Please make them more specific.
  8. Carefully read whole manuscript line by line and improve the sentence formation
  9. Cross check all references and style of reference according to Journal format, use abbreviation of journal name in reference. Reference are not matched with the number that are cited in the Manuscript such as reference 97, 98 and so on. Correct all references

The work is interesting and worth publishing.

Minor english improvement require

Round 2

Reviewer 1 Report

The author of this paper addressed all my concerns.

Please accept this current version.

Thanks

Minor editing problem

Reviewer 4 Report

The authors have not included any additional data and are relying solely on theoretical grounds by stating that the journal is interdisciplinary. Only the introduction section has been enhanced, so my concern and decision remain unchanged.

Reviewer 5 Report

Accept in present form

Minor english editing required